

# Kininogen-1 as a protein biomarker for schizophrenia through mass spectrometry and genetic association analyses

Mingjia Yang[1], Na Zhou[2], Huiping Zhang[3], Guojun Kang[1], Bonan Cao[1], Qi Kang[1], Rixin Li[1], Xiaojing Zhu[1], Wenwang Rao[1,4] and Qiong Yu[1]

[1] Department of Epidemiology and Biostatistics, School of Public Health, Jilin University, Changchun, Jilin Province, China
[2] Department of Pharmacy, Hospital of Stomatology, Jilin University, Changchun, Jilin Province, China
[3] Department of Psychiatry and Medicine (Biomedical Genetics), Boston University School of Medicine, Boston, MA, USA
[4] Unit of Psychiatry, Faculty of Health Sciences, University of Macau, Macao, China

Corresponding author
Qiong Yu, yuqiong@jlu.edu.cn

## ABSTRACT

**Background:** Schizophrenia (SCZ) is a complex and severe mental illness. There is a lack of effective biomarkers for SCZ diagnosis. The aim of this study was to explore the possibility of using serum peptides for the diagnosis of SCZ as well as analyze the association of variants in genes coding for these peptides and SCZ.
**Methods:** After bead-based fractionation, the matrix-assisted laser desorption ionization/time-of-flight mass spectrometry technique was used to identify peptides that showed different expressions between 166 SCZ patients and 201 healthy controls. Differentially expressed peptides were verified in a second set of samples (81 SCZ patients and 103 healthy controls). The association of SCZ and three tagSNPs selected in genes coding for differentially expressed peptides was performed in 1,126 SCZ patients and 1,168 controls.
**Results:** The expression level of peptides with m/z 1,945.07 was significant lower in SCZ patients than in healthy controls ($P < 0.000001$). The peptide with m/z 1,945.07 was confirmed to be a fragment of Kininogen-1. In the verification tests, Kininogen-1 had a sensitivity of 95.1% and a specificity of 97.1% in SCZ prediction. Among the three tagSNPs (rs13037490, rs2983639, rs2983640) selected in the Cystatin 9 gene (*CST9*) which encodes peptides including Kininogen-1, tagSNP rs2983640 had its genotype distributions significantly different between SCZ patients and controls under different genetic models ($P < 0.05$). Haplotypes CG (rs2983639–rs2983640) and TCG (rs13037490–rs2983639–rs2983640) were significantly associated with SCZ (CG: OR = 1.21, 95% CI [1.02–1.44], $P = 0.032$; TCG: OR = 24.85, 95% CI [5.98–103.17], $P < 0.0001$).
**Conclusions:** The present study demonstrated that SCZ patients had decreased expression of Kininogen-1 and genetic variants in Kininogen-1 coding gene *CST9* were significantly associated with SCZ. The findings from both protein and genetic association studies suggest that Kininogen-1 could be a biomarker of SCZ.

## INTRODUCTION

Schizophrenia (SCZ) is a complex psychotic disorder. It is among the top 25 mental illness in the world (*Owen, Sawa & Mortensen, 2016*; *Vos et al., 2015*). Although the prevalence of SCZ is not high (0.5–1.2%) (*Huang et al., 2019*), the health and economic burden is enormous for patients and their families as well as the society (*Chong et al., 2014*). To date, the etiology and pathogenesis of SCZ are still unknown. The diagnosis of SCZ is mainly based on descriptive and observation criteria to assess SCZ symptoms described by patients and their family members or observed by medical workers. So far, no biochemical markers or screening tools are available for SCZ diagnosis (*Santa et al., 2017*). Thus, it is important to identify SCZ biomarkers with high specificity and sensitivity, especially those present in body fluids which are easily obtained.

It is well known that SCZ is a genetic disorder. Candidate gene or genome-wide association studies have been applied to identify genetic factors for SCZ. Even though genes carrying SCZ-associated variants have been identified (*Ghazaryan et al., 2016*; *Yoosefee et al., 2016*), the mechanism by which how these genetic variants influence the vulnerability of individuals to SCZ is waiting to be explored. Since protein is the implementer of gene function, proteomic techniques such as mass spectrometry (MS) have been widely used in aiding in the diagnosis and identifying biomarkers of diseases including SCZ (*Nascimento & Martins-De-Souza, 2015*).

Mass spectrometry is an important technique for proteomics analysis. It separates and determines proteins according to the difference in mass-to-charge ratios (M/E) between different ions (*Aebersold & Goodlett, 2001*). A variety of MS techniques have been used in clinical application (*Cho et al., 2015*). For example, matrix-assisted laser desorption ionization/time-of-flight mass spectrometry (MALDI-TOF/MS), a sensitive analytical technique, has been increasingly used to evaluate the expression of specific proteins and peptides to detect biomarkers of diseases including psychiatric disorders using biological samples from patients and controls (*Ding et al., 2015*; *Huang et al., 2017*; *Kelley, Perry & Bach, 2018*; *Lo, Shiea & Huang, 2016*).

It is common that genes, proteins, and environment as well as the interaction among them lead to disease vulnerability. The exploration of disease biomarkers should conduct at multiple levels (*Sokolowska et al., 2015*). In the post-genomic era, genomics and proteomics studies are still two important areas that complement each other. Even though genetic variants may influence the structure and expression of proteins, variation in genes may not necessarily result in protein expression changes. However, changes in protein expression may reflect gene transcriptional. Recently, *Comes et al. (2018)* applied the proteomics approach in the identification of blood biomarkers for psychiatry disorders. They integrated proteomic findings from the peripheral blood and the genetic loci identified from enrichment analysis based on 30 published studies. With the availability of genomic data of a big sample the proteomic approach was demonstrated to be an essential method in discovery of biomarkers for disease including SCZ.

In present study, we initially used the MALDI-TOF/MS method to compare expression levels of serum proteins between SCZ patients and control subjects. We then screened

candidate genes that code for differentially expressed proteins. Finally, we analyzed the association between polymorphisms of protein-coding gene and SCZ. Our aim was to explore etiology and diagnosis biomarkers of SCZ at both gene and protein levels.

## MATERIALS AND METHODS

### Ethics statement

Ethics approval for the study was granted by the Ethics Committee of the School of Public Health, Jilin University (2014-05-01). All participants provided written informed consent.

### Study overview

To investigate the cause and find specific laboratory diagnostic biomarkers of SCZ, we used the bead-based fractionation method and the MALDI-TOF/MS technique to compare proteomes between first onset SCZ patients and healthy controls to identify specifically expressed proteins in SCZ patients. Verifications were conducted in another group of SCZ patients and healthy controls. Then, we searched for genes coding for differentially expressed proteins based on the National Center for Biotechnology Information (NCBI) database. Case-control genetic association analysis was used to explore the association between candidate gene polymorphisms and SCZ.

### Subjects

Serum samples for protein analysis were collected from 247 first-episode SCZ patients and 304 healthy volunteers. These first-episode SCZ patients have never taken psychiatric medications and did not receive electroconvulsive therapy in the past 3 months. DNA samples for case-control genetic association study were extracted from the peripheral blood of 1,126 SCZ patients and 1,168 healthy controls. All patients were recruited from the Changchun Mental Hospital as well as all healthy volunteers from the First Hospital of Jilin University.

All patients participated in present study were diagnosed by two independent psychiatrists. All healthy controls had no mental diseases or history of serious illness, and had not taken medications for at least 2 weeks. All subjects were Chinese Han in this study.

### Peptide extraction and MALDI-TOF/MS analysis

Peptides were extracted by copper-chelated magnetic beads. Five microliter of homogeneous magnetic beads solution and 50 µl of binding solution were added into a 0.2-ml tube and mixed thoroughly. This tube was put in a magnetic bead separator (MBS) to fix the beads, and the supernatant was aspirated. Then 5 µl of serum and 20 µl of binding solution were added and mixed, and a MBS was used to fix the beads and the supernatant was aspirated again. After washing magnetic beads with 100 µl of the washing solution, the beads were collected with MBS and the bound peptides were eluted using 20 µl of the elution solution. For target preparation, one µl of mixture containing one µl of eluent and one µl of the matrix (HCCA) solution was loaded onto a 600 µm-diameter spot size 384 MTP target plate (Bruker Daltonics, Billerica, MA, USA) and left to dry.

Analysis of the processed samples was conducted by the Ultraflex[TM] III MALDI-TOF/MS instrument (Bruker Daltonics, Billerica, MA, USA) equipped with an $N_2$ 337 nm laser.

## Candidate gene and SNP selection

The NCBI database was used to select the coding gene of Kininogen-1 protein. Three tagSNPs (rs1303749, rs2983639, and rs2983640) in the candidate gene *CST9* were selected using the Haploview program (http://hapmap.ncbi.nlm.nih.gov/). The Assay designer 3.1 was used to design the primers for genotyping these three tagSNPs. The primer sequences were as follows: rs1303490: F: 5′-ACGTTGGATGAAGCAGTTCCCAGACTTAGG-3′ R: 5′-ACGTTGGATGTCTGGGAAAACTTCCCATTC-3′; rs2983639: F: 5′-ACG TTGGA TGTCCATCTGCCCCTAAGTGAG-3′ R: 5′-ACGTTGGATGACTACCT GACTATGGA CTGC-3′; and rs2983640: F: 5′-ACGTTGGATGGTGTTCAAG GCAAACTCCAC-3′ R: 5′-ACGTTGGATGTCTGAAGAGGAAATGGGTGG-3′.

## DNA extraction and SNP genotyping

DNA was extracted from the peripheral blood using the ClotBlood DNA Kit (CoWin Biosciences, Beijing, China), and its concentration was detected by an ultraviolet spectrophotometer (Beckman, Brea, CA, USA).

SNP genotyping was conducted using the Sequenom MassARRAY platform (San Diego, CA, USA). The whole process included PCR amplification, SAP reaction, PCR extension, and desalination. The conditions of PCR was set as 94 °C for 15 min to perform a hot-start, 94 °C for 20 s to denature, 56 °C for 30 s to anneal, 72 °C for 1 min for 45 cycles to extend, 72 °C for 3 min to incubate. Then, the PCR product was incubated with SAP (Sequenom, Inc., San Diego, CA, USA) at 37 °C for 40 min. After extension and desalination, the final products were analyzed by the MassARRAY software (Sequenom, San Diego, CA, USA).

## Statistical methods

The protein spectrum analysis was conducted using the FlexAnalysis 3.0 software and the ClinProTools 2.1 software. The support vector machine and k-nearest neighbor algorithms were used to perform class prediction, and after the calculation, statistical analysis results were cross-validated. For gene polymorphism analysis, demographic characteristics were compared using the $\chi^2$ analysis and rank sum tests. Hardy–Weinberg equilibrium (HWE) was examined by the Chi-square ($\chi^2$) goodness-of-fit test. The allele frequency and genotype distribution of SNPs were analyzed by logistic regression analysis. Inheritance model and haplotype analyses were performed by the SNPStats program (https://www.snpstats.net/start.htm), and the best model was selected based on the smallest Akaike information criterion (*Akaike, 1974*). Data were analyzed using SPSS version 19.0. For all analyses, $P < 0.05$ (two-tailed) was considered statistically significant.

## RESULTS

### Comparison and identification of serum peptides
#### *Peptide profiles of SCZ patients and healthy controls in training groups*

Serum peptide profiles of 166 first onset SCZ patients and 201 healthy controls were analyzed. The mean age of 166 SCZ patients (67 males and 99 females) and 201 control subjects (90 males and 111 females) was 33 ± 13 and 36 ± 13, respectively. Serum peptide

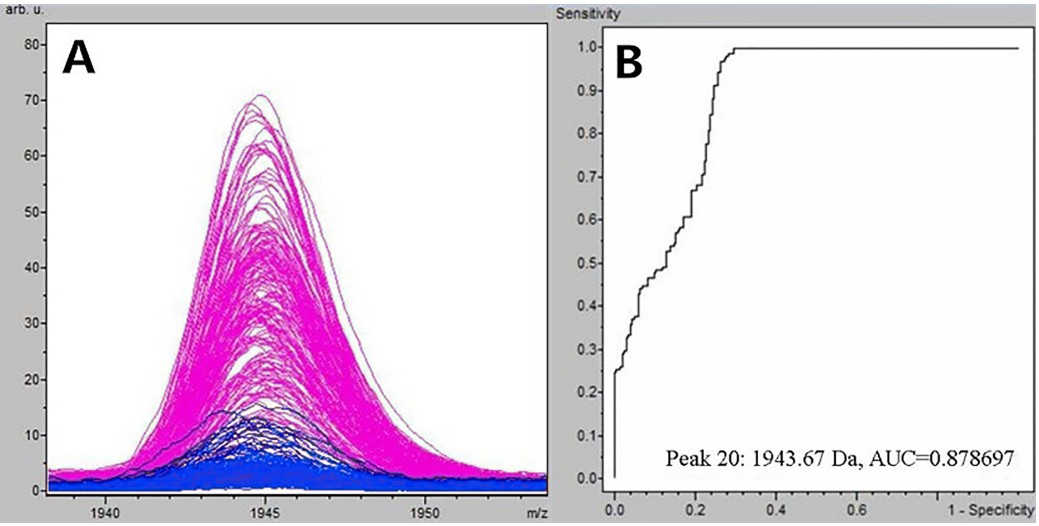

**Figure 1 Comparison of m/z 1,945.07 expression level between schizophrenia patients and healthy controls.** (A) Distribution of serum peptide with m/z 1,945.07 between schizophrenia (blue) and healthy controls (pink). (B) Receiver operating characteristic (ROC) of m/z 1,945.07 to distinguish schizophrenia patients from healthy controls.

patterns of SCZ patients were significantly different from those of healthy controls (Fig. S1).

### Selection and identification of SCZ-specific serum peptides

Through the *t*-test of ClinPro Tools software TM 2.1, with a *P*-value < 0.000001, the m/z 1,945.07 of peptides were significantly lower in patients compared to healthy controls (Fig. 1A). Using the peptide with m/z 1,945.07 to identify SCZ patients from healthy controls, the area under the curve of the receiver operating characteristic was 0.879 (Fig. 1B).

The ion of m/z 1,945.07 was confirmed to have the sequence NLGHGHKHERDQGHGHQ, which corresponded to Kininogen-1, with exactly the high molecular weight (HMW) of kininogen. The sequence map was shown in Fig. 2.

### Peptide profiles of SCZ patients and healthy controls in testing groups

Class prediction analyses were conducted in another group of SCZ patients and healthy controls. There were 81 patients (35 males and 46 females) with mean age of 32 ± 13 and 103 healthy controls (45 males and 58 females) with mean age of 35 ± 10. The spectra analysis of the validation set classified 77 of the 81 SCZ patients as SCZ positive and 100 of the 103 healthy controls as SCZ negative, revealing 95.1% sensitivity and 97.1% specificity.

## Association between *CST9* polymorphisms and SCZ
### Demographic characteristics and Hardy–Weinberg equilibrium

Among 1,076 SCZ patients, there were 591 males and 476 females with mean age of 34 ± 12. Among 1,151 healthy controls, there were 566 males and 585 females with mean age of 36 ± 10. SCZ patients and healthy controls were significantly different by age

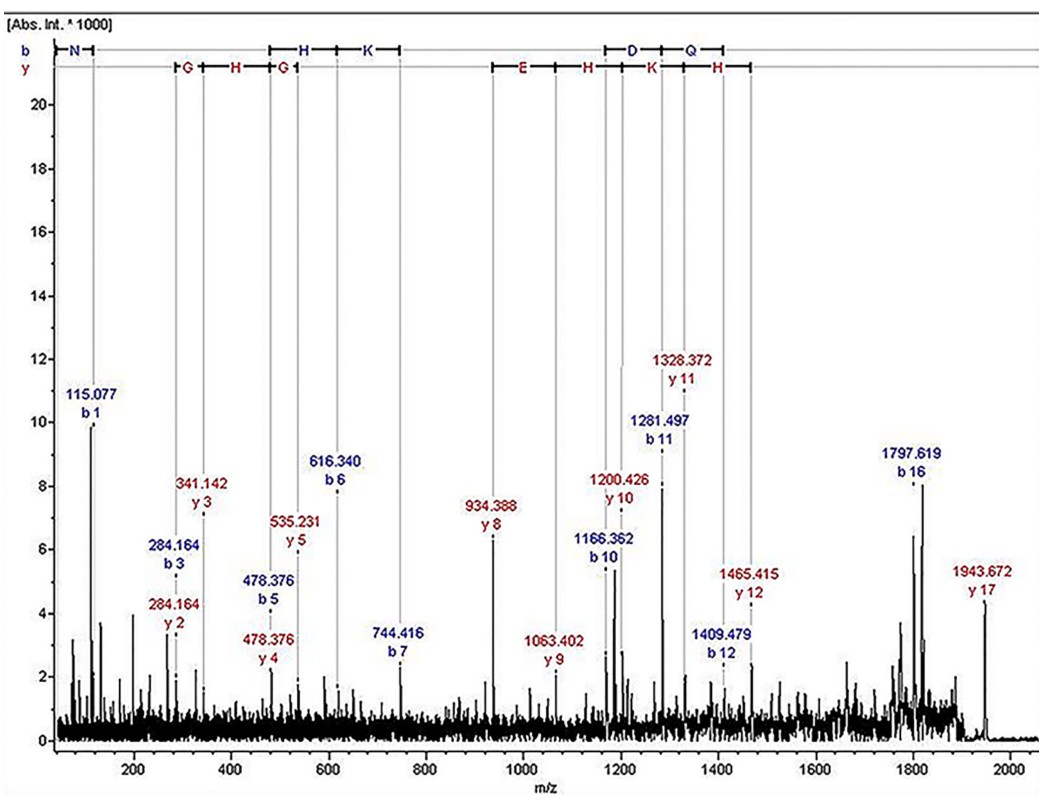

**Figure 2 The sequence map of m/z 1,945.07.**

($Z = -5.697$, $P < 0.050$) and sex ($\chi^2 = 8.569$, $P < 0.050$). Except SNP rs2983640 in the patient group, genotype distributions of the three tagSNPs (rs13037490, rs2983639, and rs2983640) all met the norm of HWE in both patient and control groups ($P > 0.050$).

### Allele frequency and genotype distribution analyses

For SNP rs2983640, the difference in genotype distributions was significantly different between patients and controls ($\chi^2 = 9.95$, $P = 0.007$), but the allele frequency was not statistically different. For rs13037490 and rs2983639, no significant differences were found in both allele frequencies and genotype distributions between patients and controls ($P > 0.05$). Details were shown in Table 1.

### Inheritance modeling analysis

For SNP rs2983640, under genetic model of codominant, recessive, or overdominant, the difference of genotype distributions between patient and control groups was significantly different (codominant: OR = 1.35, 95% CI = [1.03–1.78], $P = 0.0068$; recessive: OR = 1.45, 95% CI = [1.12–1.88], $P = 0.0051$; overdominant: OR = 0.81, 95% CI = [0.68–0.97], $P = 0.02$). However, for SNPs rs13037490 and rs2983639, under the four different genetic models, no significant difference was found between patients and controls ($P > 0.050$). The best genetic model for rs13037490 and rs2983639 was recessive and overdominant,
**Table 1 Comparison of allele frequencies and genotype distributions of three tagSNPs between schizophrenia patients and healthy controls.**

| SNP | Genotype/Allele | SCZ (%) | Control (%) | $\chi^2$ | $P$ |
|---|---|---|---|---|---|
| rs13037490 | T/T | 807 (75) | 864 (76) | 0.71 | 0.702 |
| | T/C | 245 (23) | 264 (23) | | |
| | C/C | 19 (2) | 16 (2) | | |
| | T | 1,859 (87) | 1,992 (87) | 0.13 | 0.719 |
| | C | 283 (13) | 296 (13) | | |
| rs2983639 | C/C | 682 (65) | 713 (62) | 4.19 | 0.123 |
| | C/T | 320 (30) | 393 (34) | | |
| | T/T | 49 (5) | 44 (4) | | |
| | C | 1,684 (80) | 1,819 (79) | 0.67 | 0.412 |
| | T | 418 (20) | 481 (21) | | |
| rs2983640 | A/A | 424 (44) | 492 (43) | 9.95 | 0.007 |
| | A/G | 398 (41) | 528 (46) | | |
| | G/G | 143 (15) | 124 (11) | | |
| | A | 1,246 (65) | 1,512 (66) | 1.13 | 0.287 |
| | G | 684 (35) | 776 (34) | | |

Note:
$P$-values were adjusted for age and gender.

**Table 2 Comparison of inherence models between patients and controls.**

| SNP | Model | Genotype | SCZ (%) | Control (%) | OR [95% CI] | $P$ | AIC |
|---|---|---|---|---|---|---|---|
| rs13037490 | Recessive | T/T–T/C | 1,024 (98.2) | 1,128 (98.6) | 1.00 | 0.4 | 3,026.5 |
| | | C/C | 19 (1.8) | 16 (1.4) | 1.34 [0.68–2.62] | | |
| rs2983639 | Overdominant | C/C–T/T | 724 (69.5) | 757 (65.8) | 1.00 | 0.058 | 3,005.8 |
| | | C/T | 317 (30.4) | 393 (34.2) | 0.84 [0.70–1.01] | | |
| rs2983640 | Codominant | A/A | 419 (43.8) | 492 (43) | 1.00 | 0.0068 | 2,864.5 |
| | | G/A | 394 (41.2) | 528 (46.1) | 0.87 [0.72–1.05] | | |
| | | G/G | 143 (15) | 124 (10.8) | 1.35 [1.03–1.78] | | |
| | Recessive | A/A–G/A | 813 (85) | 1,020 (89.2) | 1.00 | 0.0051 | 2,864.7 |
| | | G/G | 143 (15) | 124 (10.8) | 1.45 [1.12–1.88] | | |
| | Overdominant | A/A–G/G | 562 (58.8) | 616 (53.9) | 1.00 | 0.02 | 2,867.1 |
| | | G/A | 394 (41.2) | 528 (46.1) | 0.81 [0.68–0.97] | | |

Note:
OR (95% CI) and $P$-values were adjusted for age and gender.

respectively. The best genetic model for rs13037490 and rs2983639 as well as the results from the difference models for rs2983640 were listed in Table 2.

### Haplotype analysis

When haplotypes with frequency of more than 0.01 being considered, haplotypes CG (rs2983639–rs2983640) and TCG (rs13037490–rs2983639–rs2983640) were significantly associated with SCZ (CG: OR = 1.21, 95% CI [1.02–1.44], $P$ = 0.032; TCG: OR = 24.85, 95% CI [5.98–103.17], $P$ < 0.0001). No other haplotypes showed significant associations with SCZ.
## DISCUSSION

Even though there have been some studies reported that Kininogen-1 was significantly reduced in SCZ patients, our study was conducted in northern Han Chinese population, and we explored if variants in its coding gene were associated with SCZ. We confirmed that Kininogen-1 was a serum protein biomarker of SCZ, and it was lowly expressed in SCZ patients. SNP rs2983640 in *CST9*, one of the coding genes of Kininogen-1, was significantly associated with SCZ. Additionally, the frequency of haplotypes carrying *CST9* variants was significantly different between SCZ patients and healthy controls.

There are two types of Kininogens, low molecular weight kininogen and HMW kininogen (*Lalmanach et al., 2010*), in mammals. In 1999, Irina firstly reported the association between SCZ and the kallikrein–kinin system (KKS). He found that plasma KKS system activity was increased in SCZ patients, and kallikrein in the brain of patients was significantly reduced. This finding suggested that KKS might play a role in blood–brain barrier and brain tissue damage, leading to the development of SCZ (*Shcherbakova et al., 1999*). Kininogen-1 is an important component of KKS. Together with factor XII, kallikreins, kinins, and kininases, it is involved in the regulation of multiple organs and diverse pathophysiological processes such as cardiovascular, renal, nervous system, tumor, thrombosis, atherosclerosis and cell proliferation, inflammation, and apoptosis (*Albert-Weissenberger, Siren & Kleinschnitz, 2013*; *Puchades et al., 2003*; *Regoli & Gobeil, 2015*; *Rhaleb, Yang & Carretero, 2011*).

High molecular weight is a multifunctional glycoprotein in the plasma. It is synthesized by hepatocytes, and can be degraded to bradykinin (BK) under the action of kallikrein. BK receptors are also distributed in the central nervous system and play a role in diseases of the central nervous system (*Chen et al., 2000*). BK induces the synthesis and release of other inflammatory mediators in the central nervous system and the glial tissue, causing brain tissue damages or prolonged disturbance of blood–brain barrier function, which in turn can lead to brain damage (*Ding-Zhou et al., 2002*).

The production of kinins is influenced by the expression of kininogen. There was evidence about the mediatory role of kinins in the inflammatory response associated with different neurological disorders (*Guevara-Lora, 2012*). In neurological diseases, such as Alzheimer's disease, Parkinson's disease, and multiple sclerosis, proteins involved in kinin generation or kinin receptor function were overexpressed (*Khandelwal, Herman & Moussa, 2011*; *Lisak et al., 2011*).

Kininogen is a member of the cystatin superfamily, which consists of type 1 cystatins (stefins), type 2 cystatins, and the kininogens. The results of this study indicated that compared with healthy controls, HMW was significantly reduced in SCZ patients. The deficiency of kininogen may lead the balance between proteases and protease inhibitors to be broken down in SCZ patients, resulting in the increase of homocysteine (Hcy) levels in the serum. Since Hcy is neurotoxic can affect the development of the central nervous system, it may be involved in the pathogenesis of SCZ. Several studies have shown that Hcy is significantly higher in SCZ patients than in healthy controls (*Kale et al., 2010*; *Moustafa et al., 2014*; *Numata et al., 2015*).

Kininogen is coded by the cystatin 9 gene (*CST9*), and cystatin 9 belongs to the Cystatins superfamily. It is one of 14 members of type 2 cystatins, which were found in human body compartments and fluids (*Ochieng & Chaudhuri, 2010*). In this study, we explored the association between three tagSNPs in *CST9* and SCZ. Since age and gender of patients and controls were not well matched, we adjusted their confounding effect in the analysis. In the HWE test, SNP rs2983640 in the patient group did not meet the norm of HWE. This may be due to that rs2983640 is associated with SCZ. Findings from previous studies also indicate that SNPs that are not in HWE may contribute to complex diseases (*Mayo, 2008*). For rs2983640, the genotype distribution was significantly different between patients and controls under the genetic model of codominant, recessive, or overdominant. Haplotypes CG (rs2983639–rs2983640) and TCG (rs13037490–rs2983639–rs2983640) were found to be associated with SCZ. All these results imply that polymorphisms of *CST9* were associated with the risk of SCZ. Our genetic association studies not only indicated the association between *CST9* variants and SCZ, but also confirmed that the Kininogen-1 protein could be a biomarker for the diagnosis of SCZ.

However, there is still follow-up work waiting to be done in the future. For example, we need to conduct the study with a larger sample to ensure having sufficient statistical power. Subjects in this study were recruited in the northeast of China, and they were all Han Chinese. We need to conduct studies in subjects from other areas or ethnicity groups. We also need to study the effect of *CST9* interaction with other genetic or environmental factors on the occurrence of SCZ. Finally, candidate biomarkers should be validated in the absence of interference from other diseases.

## CONCLUSIONS

In summary, our study showed that using the MALDI-TOF/MS technique, Kininogen-1 could be a biomarker to distinguish SCZ patients from healthy controls with a high sensitivity and specificity. We also confirmed this at the gene level, that is, variants in CST9, the coding gene of Kininogen-1, were associated with the risk of SCZ. Both protein and genetic association studies demonstrated that Kininogen-1 could be a biomarker of SCZ. Definitely, our findings should be validated in a larger sample and replicated in different populations.

## ACKNOWLEDGEMENTS

We are grateful to Steven Limbara for checking the language of this manuscript.

### Funding

This work was supported by the Natural Science Foundation of China (grant no. 81673253), the Norman Bethune Program of Jilin University (2015227), the Projects of International Cooperation and Exchanges NSFC (no. 81320108025), and the Graduate Innovation Fund of Jilin University (2017161). The funders had no role in study design, data collection and analysis, decision to publish, or preparation of the manuscript.

## Grant Disclosures

The following grant information was disclosed by the authors:

Natural Science Foundation of China: 81673253.

Norman Bethune Program of Jilin University: 2015227.

Projects of International Cooperation and Exchanges NSFC: 81320108025.

Graduate Innovation Fund of Jilin University: 2017161.

## Competing Interests

The authors declare that they have no competing interests.

## Author Contributions

- Mingjia Yang conceived and designed the experiments, performed the experiments, analyzed the data, authored or reviewed drafts of the paper, approved the final draft.
- Na Zhou conceived and designed the experiments, performed the experiments, analyzed the data, authored or reviewed drafts of the paper, approved the final draft.
- Huiping Zhang performed the experiments, analyzed the data, authored or reviewed drafts of the paper, approved the final draft.
- Guojun Kang performed the experiments, analyzed the data, authored or reviewed drafts of the paper, approved the final draft.
- Bonan Cao analyzed the data, authored or reviewed drafts of the paper, approved the final draft.
- Qi Kang contributed reagents/materials/analysis tools, authored or reviewed drafts of the paper, approved the final draft.
- Rixin Li contributed reagents/materials/analysis tools, authored or reviewed drafts of the paper, approved the final draft.
- Xiaojing Zhu prepared figures and/or tables, authored or reviewed drafts of the paper, approved the final draft.
- Wenwang Rao prepared figures and/or tables, authored or reviewed drafts of the paper, approved the final draft.
- Qiong Yu conceived and designed the experiments, authored or reviewed drafts of the paper, approved the final draft.

## Human Ethics

The following information was supplied relating to ethical approvals (i.e., approving body and any reference numbers):

Ethics approval for the study was granted by the Ethics Committee of the School of Public Health, Jilin University (2014-05-01).

## Data Availability

Raw data is available as a Supplemental File.

## Supplemental Information

Supplemental information for this article can be found online at http://dx.doi.org/10.7717/peerj.7327#supplemental-information.

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
