# Peer review of "Kininogen-1 as a protein biomarker for schizophrenia through mass spectrometry and genetic association analyses"

_PeerJ, doi:10.7717/peerj.7327_

## Round 0.1 · original submission · Major Revisions

This is an interesting article on the bio-marker of schizophrenia. However, it still has some issues need to be addressed. I suggest the authors to consider the comments provided by the two reviewers and revise their paper carefully.

Reviewer 1 ·

Basic reporting

The paper needs reorganization in some sections to make it clearer and easy to understand.

Experimental design

There are some additional experiments that need to be added in this manuscript.

Validity of the findings

The finding is meaning but needs to be further confirmed.

Additional comments

General Comments: Schizophrenia is a mental illness, which serious influence the life of human beings. However, there is still no effective testing methods for its clinical diagnosis. In this manuscript, the authors have explored the application of using Kininogen-1 as a protein biomarker for schizophrenia diagnosis, which is meaningful for the early diagnosis of schizophrenia and its prediction. However, there are some big problems in this manuscript, and it can be acceptable after major revision. The details have been listed below:

1, The authors claimed that it was the first study that explored if Kininogen-1 could be a protein biomarker for the diagnosis of schizophrenia. However, there are some studies have already reported that Kininogen-1 was significantly reduced in SCZ patients. The novelty of this study should be further described.

2, As a new testing method for the clinical diagnosis of schizophrenia, it should be compared with the traditional method to investigate its reliability. I understand that the receiver operating characteristic has been used to confirm this. But more experiments need to be carried out including the double-blind experiments.

3, Figure 1 showed that the expression level of Kininogen-1 was obvious decreased in schizophrenia patients compared with healthy controls. Would the expression level of Kininogen-1 different from each other for the schizophrenia patients under severe, moderate and mild level? These data should be provided in this manuscript.

4, In this study, the authors explored the association between three tagSNPs in CST9 and SCZ. The results should be further confirmed by real-time PCR method.

5, The resolution of Figure 1 is too low. Both these figures should be made by Excel, origin or other software but not directly take a photo and put in the manuscript.

6, There are some language errors in this manuscript. The whole manuscript needs to be edited by an English-speaking author. The paper needs reorganization in some sections.

Annotated reviews are not available for download in order to protect the identity of reviewers who chose to remain anonymous.

Reviewer 2 ·

Basic reporting

no comment

Experimental design

No comment

Validity of the findings

No comment

Additional comments

This is a pretty interesting and meaningful study exploring whether Kininogen-1 protein is a biomarker for schizophrenia diagnosis and the relationship between protein-coding gene (CST9) and schizophrenia. However, some issues should be solved. My detailed comments are as follows:

Title: - The title is not very precise.
Abstract:
-In this part, the authors should clearly tell us what the specific protein-coding gene is?
-The results are not perfectly summarized.
-The conclusion is a bit simple.

Introduction:
-Line 40. Reference 3 is out of date. Authors should cite the latest literatures published in the Lancet or its sub-journals.
-The authors should introduce the latest schizophrenia-associated proteomics and genomic studies to highlight the importance of current research.
-Page 67. “we first adopted MALDI-TOF/MS to profile serum protein patterns in SCZ patients and control subjects”. This sentence is an inappropriate expression. Please correct it.
-Page 69-70. The authors should use “protein-coding gene” rather than “candidate genes”.Method:-Page 94-95. Both the subheadings of Ethics statement and subheadings of Subjects mentioned participants from their study wrote informed consent. Please delete one of them.
-Page 111. Please provide the full names of CST 9, when it appears in the Abstract or mai text for the first time.
-Page 135-136. Authors mentioned that P values were adjusted for age and gender in Table 1 and Table 2. However, in the statistic section, authors mentioned that the allele frequency and genotype distribution of SNPs were analyzed by χ 2 tests. As far as I know, Chi-square test cannot implement controlling variables. Please introduce the rationale clearly.-Page 138. please add references about AIC.

Results:
-Page 146-147. There are not any statistical figures and graphics to prove this result. Please provide more information.
-Page 162. Due to the fact that CST 9 appears in the main text for the second time, an abbreviated form will be more concise than the full name.

Discussions:
-Page 241-249. A complete paragraph about research limitations important for readers. Thus, it is not common to combine conclusions with limitations.-Gene CST 9 interacted with other genetic (or environmental) factors, may play a role in the occurrence of SCZ. Please add this limitation.

Conclusions:-Further research content can be appropriately described and forecast in this section.

---

## Round 0.2 · Minor Revisions

Minor revision is required. Please revise the paper accordingly. Before re-submission, please carefully check the language and make necessary edits.

Reviewer 1 ·

Basic reporting

Good.

Experimental design

The experiments are well-designed.

Validity of the findings

The finding is well investigated.

Additional comments

General Comments: The authors have developed a novel method, which uses Kininogen-1 as a protein biomarker for schizophrenia diagnosis. Authors have addressed some concerns but there are still some problems. Therefore, the manuscript can be acceptable after minor revision. The details have been listed below:

1, The resolution of Figure 1 is still too low to recognize. Authors should describe the X,Y axes in Figure 1A. And the sensitivity and specificity are very unclear in Fig. 1B. So it is highly recommended to increase the quality of this figure because it is one of the most important figure in this work.

2, In the response, authors said that “the mRNA expression level of CST9 gene was too low to reach the detection threshold, so we failed to detect its mRNA expression level and failed to compare the different of mRNA expression level between patients and controls”. I understand that mRNA is in low level and unstable, but after reversed to cDNA and combined with qPCR, the difference of mRNA expression level between patients and controls maybe detectable.

Reviewer 2 ·

Basic reporting

pass

Experimental design

pass

Validity of the findings

pass

Additional comments

The authors have made good responses to the comments/suggestions of this reviewer. No more revision is needed.

---

## Round 0.3 · accepted · Accept

I am satisfied with your revisions. Thank you.